# Universal Single-Dose Vaccination against Hepatitis A in Children in a Region of High Endemicity

**DOI:** 10.3390/vaccines8040780

**Published:** 2020-12-20

**Authors:** Mikhail I. Mikhailov, Maria A. Lopatukhina, Fedor A. Asadi Mobarhan, Lyudmila Yu. Ilchenko, Tatyana V. Kozhanova, Olga V. Isaeva, Anastasiya A. Karlsen, Ilya A. Potemkin, Vera S. Kichatova, Anna A. Saryglar, Natalia D. Oorzhak, Karen K. Kyuregyan

**Affiliations:** 1Department of Viral Hepatitis, Russian Medical Academy of Continuous Professional Education, 125993 Moscow, Russia; michmich2@yandex.ru (M.I.M.); isaeva.06@mail.ru (O.V.I.); karlsen12@gmail.com (A.A.K.); axi0ma@mail.ru (I.A.P.); vera_kichatova@mail.ru (V.S.K.); 2Laboratory of Viral Hepatitis, Mechnikov Research Institute for Vaccines and Sera, 105064 Moscow, Russia; m.lopatukhina@gmail.com (M.A.L.); 1amfa@bk.ru (F.A.A.M.); ilchenko-med@yandex.ru (L.Y.I.); 3Chumakov Federal Scientific Center for Research and Development of Immunobiological Products of Russian Academy of Sciences, 108819 Moscow, Russia; vkozhanov@bk.ru; 4Kyzyl Hospital of Infectious Diseases, 667003 Kyzyl, Russia; anna_kyzyl@mail.ru; 5Tuva Regional Service for Surveillance, 667010 Kyzyl, Russia; natalia.oorzhak@yandex.ru

**Keywords:** hepatitis A, hepatitis A vaccine, single-dose vaccination, epidemiology, incidence, public health

## Abstract

Since August 2012, universal single-dose vaccination in children aged at least three years has been implemented in the Republic of Tuva, which was previously the region most affected by hepatitis A in Russia. The objective of this cross-sectional study was the assessment of the immunological and epidemiological effectiveness of vaccination program five years following its implementation. In the pre-vaccination period, anti-HAV antibody detection rates in Tuva was 66.0% [95% CI: 56.3–74.6%] in children aged 10–14 years and reached a plateau (>95%) by age 20–29 years. Annual incidence rates in children under 18 years of age peaked at 450–860 per 100,000 in pre-vaccination years but dropped to 7.5 per 100,000 in this age group and to 3.2 per 100,000 in the total population one year after the start of vaccination. Since 2016, no cases of hepatitis A has been reported in Tuva. Serum anti-HAV antibodies were quantified in samples from healthy children following single-dose vaccination. Protective anti-HAV antibody concentrations (≥10 mIU/mL) were detected in 98.0% (95% CI: 96.2–99.0% (442/451)) of children tested one month after single-dose immunization, in 93.5% (95% CI: 91.0–95.4% (477/510)) and in 91.1% (95% CI: 88.2–93.4% (422/463)) of children one year and five years after single-dose immunization, respectively. Anti-HAV antibody geometric mean concentrations were similar in sera collected one month, one year, and five years following single-dose vaccination: 40.24 mIU/mL, 44.96 mIU/mL, and 57.73 mIU/mL, respectively (*p* > 0.05). These data confirm that single-dose vaccination is an effective method of bringing hepatitis A under control in a short period of time in a highly endemic region.

## 1. Introduction

Hepatitis A is a disease preventable by vaccination. There are two types of vaccines against hepatitis A—live attenuated vaccines, adopted in China [1], and inactivated vaccines that are available worldwide [2]. Targeted immunization of at-risk populations (sewage treatment plant workers, homeless individuals, intravenous drug users, and men who have sex with men) is effective in preventing sporadic cases and small-scale outbreaks, but does not lead to the formation of herd immunity, and therefore, is not effective in terms of reducing the incidence of hepatitis A in the general population. Universal pediatric vaccination programs may be beneficial in regions with endemicity transition from high to intermediate. In such regions, the circulation of the hepatitis A virus (HAV) among children decreases due to the improved sanitation. As a result, however, the proportion of susceptible adolescents and adults increases. As children typically exhibit asymptomatic hepatitis A infection, this leads to an overall increase in the number of clinically significant and severe cases of the disease [3]. For countries experiencing this kind of epidemiological shift, where the disease burden is greatest, the World Health Organization (WHO) recommends implementation of universal vaccination programs [4]. Several countries introduced universal two-dose vaccination against hepatitis A in national immunization schedules for children aged ≥1 year (Israel, Panama, Uruguay, and 17 states of the USA). This resulted in a rapid 93–98% decline in disease incidence [2].

The standard immunization schedule for the inactivated vaccine against hepatitis A consists of two doses given six months apart. This vaccination schedule proved to be highly immunogenic and provides a protective antibody response which is expected to last for decades [5]. Moreover, protective anti-HAV antibody levels can persist for at least 11 years after a single dose of the inactivated hepatitis A vaccine and can also increase or reappear after a booster vaccination [6]. For economic reasons and to improve coverage rates, the universal single-dose vaccination against hepatitis A was first implemented in toddlers in Argentina in 2005 [7], and then in Brazil in 2014 [8]. Furthermore, single-dose vaccination was also implemented by the Republic of Korea Armed Forces in 2013 [9]. These immunization programs have proven to be effective both in terms of immunological response and in terms of impact on reported disease incidence. However, more data from different regions with varying epidemiological conditions are needed to understand its effectiveness at the general population level. Based on the experience of Argentina and in conjunction with the WHO recommendation [4], universal pediatric single-dose vaccination has been implemented since August 2012 in the Tuva Republic, which was the region most severely affected by hepatitis A in the Russian Federation. Tuva is located in southern Siberia and borders Mongolia to the south. According to the United Nations’ Human Development Index (HDI), Tuva is the least developed region in Russia [10]. Hepatitis A incidence in children under 17 years in Tuva was consistently 10–30 times higher than the average incidence in Russia in the pre-vaccination period, until 2013. The vaccination campaign in Tuva was initiated in August 2012 by the Chumakov Institute of Poliomyelitis and Viral Encephalitides based in Moscow, Russia. Since then, single-dose vaccination against hepatitis A for children over three years of age has been introduced into the Tuva regional immunization schedule.

The objective of our study was the assessment of the epidemiological and immunological effectiveness of single-dose vaccination against hepatitis A over a five-year period in the Republic of Tuva. We also present the data on the herd immunity to HAV in Tuva in pre-vaccination period to give a more comprehensive epidemiological characterization of the study region.

## 2. Materials and Methods

### 2.1. Incidence Analysis

Data on the annual incidence of hepatitis A were retrieved from the database of the Russian Federal Service for Surveillance on Consumer Rights Protection and Human Wellbeing (Rospotrebnadzor). Annual incidence rates in Tuva in 2001–2019 were compared to the average rate for the entire Russian Federation. Incidence rates registered in Tuva in 2013–2019 were also compared to those observed in the pre-vaccination period, both for the total population and for children under 14 years.

Confirmed hepatitis A cases registered after 2012 were investigated retrospectively by reviewing the database of the Tuva regional office of Rospotrebnadzor. The criteria for hepatitis A cases were as follows: (a) IgM anti-HAV positive and (b) jaundice and/or elevated liver enzymes with acute onset. Retrospective analysis included demographic data, history of hepatitis A vaccination, clinical data, and infection risk factors, including contacts with infected patients and travel outside the region.

### 2.2. Study Cohorts

To assess the herd immunity to HAV in the pre-vaccination period, antibodies for HAV (anti-HAV) IgG were tested in archived serum samples obtained in 2008 from 1011 conditionally healthy individuals from Tuva. Ten age groups were enrolled: less than 1 year, 1–4 years, 5–9 years, 10–14 years, 15–19 years, 20–29 years, 30–39 years, 40–49 years, 50–59 years, and over 60 years; each age group included roughly 100 people (a minimum of 88, a maximum of 123). All participants or their legal guardians completed questionnaires containing questions about age, gender, social conditions, sources of water supply, and hepatitis A vaccination, and gave informed consent for the research. The male-to-female ratio was 1:1.8 and the rural-to-urban population ratio was 1.7:1. These samples were originally obtained for the purpose of studying the prevalence of hepatitis viruses and the development of herd immunity to these infections in the Russian Federation. The study design was approved by the Ethics Committee of the Chumakov Institute of Poliomyelitis and Viral Encephalitides in Moscow, Russia (Approval #6 dated 2010-04-01). All samples were stored in aliquots at −70 °C until testing.

Serum anti-HAV antibodies were quantified in samples from healthy children who were vaccinated against hepatitis A under the single-dose schedule. Three different cohorts of children were surveyed at three separate time points: (1) for 451 children (aged 3–8 years, median age 6 years), samples were collected in 2012, one month after vaccination; (2) for 510 children (aged 4–8 years, median age 6 years), samples were collected in 2013, one year after vaccination; and (3) for 463 children (aged 7–13 years, median age 10 years), samples were collected in 2017–2018, five years after vaccination. Within the cohorts of children in the study, the male-to-female ratio ranged from 1:0.8 to 1:1.5, and the rural-to-urban population ratio ranged from 1:4 to 1:5.

The study was conducted in accordance with the principles expressed in the World Medical Association Declaration of Helsinki regarding ethical medical research involving human subjects. Written informed consent was obtained from the parents (or legal guardians) of all participants. The study design was approved by the Ethics Committee of the Chumakov Institute of Poliomyelitis and Viral Encephalitides in Moscow, Russia (Approval #4 dated 2012-04-01). The volunteers were persons undergoing routine medical examinations, visitors to the vaccination office undergoing routine vaccinations, and patients visiting the polyclinic for reasons not related to infectious diseases. Inclusion criteria were a history of single-dose vaccination against hepatitis A with available vaccination records, and a signed and dated informed consent form approved by the Ethics Committee. Exclusion criteria were as follows: children in care, vaccination with two doses of the HAV vaccine, and subjects with a known prior history of HAV infection before vaccination.

Blood samples 3 mL in volume were obtained from each participant. All sera samples were coded and aliquoted, and aliquots were stored at −70 °C until testing.

### 2.3. Anti-HAV Testing

Total anti-HAV antibodies were tested in the sera of vaccinated children using quantitative immunoassay Elecsys^®^ Anti-HAV (Roche, Mannheim, Germany) on a Cobas e411 analyzer according to the manufacturer’s instructions. Seropositivity was defined as antibody levels of ≥10 mIU/mL, as the reported minimal serum levels of total anti-HAV antibodies required for protection against HAV in humans is 10 mIU/mL [11]. All samples with anti-HAV concentrations above the upper limit of the quantification range of the assay were diluted and repeatedly tested. The final concentrations were obtained by multiplying the result by the dilution factor. Children with antibody levels of <10 mIU/mL were offered a second dose of the vaccine. No additional measurement of antibodies was performed thereafter for these children.

Total anti-HAV IgG antibodies were tested in archived serum samples from conditionally healthy individuals using the Enzyme-Linked Immunosorbent Assay (ELISA) Monolisa Total Anti-HAV Plus Kit (BioRad, Coquette, France) according to the manufacturer’s specifications. Samples were considered positive when the concentration of anti-HAV antibodies measured ≥20 mIU/mL, according to the kit manual.

### 2.4. Statistical Analysis

Data analysis was performed using graphpad.com. Statistical analysis included geometric mean concentration (GMC) for anti-HAV antibody concentrations, the calculation of a 95% confidence interval (95% CI), and assessing the significance of differences of mean values between groups using Fisher’s exact test (significance threshold *p* < 0.05).

## 3. Results

### 3.1. Analysis of Hepatitis A Incidence

The vaccination campaign in Tuva was initiated in August 2012 with the monovalent pediatric inactivated vaccine (HAVRIX^®^ 720 EU) given to children aged 3–8 years. By the end of 2012, a total of 65,097 children had received single-dose immunization, resulting in 87.4% coverage in children aged 3–8 years.

Registered incidence of infectious diseases in Russia is reported by Rospotrebnadzor for three categories of people: the total population, children under 15 years of age, and children under 18 years of age. Hepatitis A incidence in Tuva in the pre-vaccination period (2001–2012) was highest in the final category, reaching 71.0–869.5 per 100,000 in children 0–17 years, compared to the overall average in Russia of 7.5–183.1 per 100,000 (Figure 1).

In 2013, the year immediately following the implementation of the single-dose vaccination program, the hepatitis A incidence in Tuva dropped to 7.5 per 100,000 in children aged under 18 years and continued to decrease in subsequent years, until the incidence rates reached zero in 2016–2019. Figure 2 shows hepatitis A incidence in Tuva from 2012 to 2019 in greater detail, both in the total population and in children aged under 18 years, i.e., the age cohort that included vaccinated children.

In 2013, hepatitis A incidence dropped 96.9% (to 7.5 per 100,000) in children aged 0–18 years and 96.7% (to 3.2 per 100,000) in the total population. In 2014, hepatitis A incidence in Tuva decreased further to 2.3 per 100,000 in the total population and 4.6 per 100,000 in children aged 0–18 years. Only 19 cases of hepatitis A have been reported in Tuva since the start of the vaccination campaign, including 13 cases in children. A detailed description of all cases reported in Tuva in 2013–2015 is shown in Table 1. All pediatric cases were reported in unvaccinated children. Three of these pediatric cases, along with one adult case, were determined to have been imported from neighboring Kyrgyzstan, where it was confirmed the subjects came into contact with an HAV-positive individual. In 2015, only two cases of hepatitis A were registered, including one case in a child under 14 years. In 2016–2019, no cases of hepatitis A were registered in Tuva, making it the only region in Russia that was free of hepatitis A incidence.

### 3.2. Herd Immunity to HAV in TUVA in the Pre-Vaccination Period

The average anti-HAV IgG positivity rate in the general population in Tuva in 2008, during the pre-vaccination period, was 77.3% (782/1,011, 95% CI: 74.7–79.8%). The proportion of seropositive individuals was lowest in children in the 1–4 years, 5–9 years, and 10–14 years age groups, then increased in adolescents to 86% and reached a plateau (>95%) by age 20–29 years (Table 2). Thus, a 50% level of anti-HAV antibody detection in Tuva was reached in 2008 for the group of children aged 10–14 years. More than 50% of tested children aged under one year had anti-HAV IgG, suggesting the presence of maternal antibodies. The proportion of vaccinated subjects in the tested cohort was very small (4.6%), reaching a maximum of only 15% of children in the 5–9 years age group.

The frequency of anti-HAV antibody detection among the rural population was significantly higher than among the urban population (81.6% (515/631) vs. 70.1% (267/380), *p* < 0.01). Furthermore, anti-HAV was detected in women at a higher rate when compared to men (81.5% (527/647) vs. 70.1% (255/364), *p* < 0.01). According to the questionnaire data, only 52.6% (532/1011) of surveyed participants had access to tap water as a source of drinking water, while others utilized wells or open reservoirs as their principal water supply. The anti-HAV seropositivity rates were similar in those who used tap water and in individuals who had other sources of water supply (wells, open reservoirs) (77.4% (412/532) vs. 77.2% (370/479), *p* > 0.05).

### 3.3. Persistence of Anti-HAV Antibodies Following Single-Dose Vaccination

Protective anti-HAV antibody concentrations (≥10 mIU/mL) were detected in 98.0% (95% CI: 96.2–99.0% (442/451)) of children tested one month after single-dose immunization, in 93.5% (95% CI: 91.0–95.4% (477/510)) and in 91.1% (95% CI: 88.2–93.4% (422/463)) of children one year and five years after single-dose immunization, respectively. The difference in rates of seroprotection between cohorts surveyed one year and five years following vaccination were not statistically significant (*p* > 0.05, Fisher’s exact test), although the observed decrease in the rate of seroprotection compared to the cohort surveyed one month after the vaccination was statistically significant (*p* < 0.01, Fisher’s exact test).

All reactive samples obtained both one month and one year following immunization contained anti-HAV antibodies in concentrations within the range 20 to 6000 mIU/mL. The majority of reactive samples in the cohort of samples tested five years following single-dose immunization contained anti-HAV antibody concentrations within the range 20 to 6000 mIU/mL (Table 3), although the proportion was significantly lower compared to cohorts surveyed one month and one year following single-dose vaccination (*p* < 0.01, Fisher’s exact test). In the cohort surveyed five years following vaccination, 20 serum samples containing anti-HAV antibodies in concentrations above 6000 mIU/mL were identified, suggesting a “boost” following exposure to HAV or undocumented repeat vaccination.

Anti-HAV antibody GMCs were similar in serum samples collected one month, one year, and five years following single-dose vaccination (*p* > 0.05, Fisher’s exact test) when samples with anti-HAV antibody concentrations <10 mIU/mL and >6000 mIU/mL were excluded from the calculation of GMC (Table 4).

## 4. Discussion

Single-dose vaccination against hepatitis A, first implemented in Argentina, may be a simple and cost-effective solution compared to the standard immunization schedule when maximum vaccination coverage needs to be achieved quickly. The first randomized trial of a single-dose vaccination schedule with an inactivated HAV vaccine was done in 2003 in Nicaragua. The protective efficacy was demonstrated to be 85% within 6 weeks and 100% after 6 weeks following the immunization [12].

Reports from Argentina following the implementation of this approach demonstrated a rapid decrease in disease incidence, both in vaccinated children and in the total population, proving that it is a cost-effective way of quickly maximizing the vaccination coverage and reducing disease [13,14]. Following the success of the vaccination program in Argentina, this novel strategy was implemented in 2012 in Tuva, where hepatitis A incidence rates significantly exceeded those observed in Argentina and Brazil (which implemented single-dose vaccination in 2014) before either country implemented their respective single-dose vaccination programs. Indeed, peak pre-vaccination incidence rates in Tuva reached 400 to 850 per 100,000 in children aged under 18 years compared to 66.5 per 100,000 reported in Argentina [7] and 10–11 cases per 100,000 in children aged 5–14 years in Brazil [8]. These incidence rates, together with data from our seroprevalence study, prove that Tuva was highly endemic for hepatitis A for a long period of time. In a cohort sampled in 2008, almost 100% of healthy individuals over the age of 20 years were seropositive, suggesting long-term intensive HAV circulation in this region. The relatively low seroprotection rates observed during the pre-vaccination period in children aged 1–14 years may explain why the majority of cases reported in Tuva were reported in children and adolescents.

The majority of universal HAV vaccination programs are focused on toddlers [2], although in some regions it may be targeted to older cohorts, depending on the epidemiology. For example, in Catalonia (Spain) universal hepatitis A + B vaccination was introduced for 12-year-olds [15]. In the case of Tuva, the vaccination program targets children aged three years and older. This was intended to make all HAV vaccines approved in Russia available for use in this vaccination program. While Havrix is approved for use in children aged 12 months and older, some vaccines, including one produced in the Russian Federation, are approved only for children aged at least three years.

Following the start of vaccination, a rapid decrease in hepatitis A incidence was observed in Tuva, not only in vaccinated children, but also in non-vaccinated adolescents and adults. This may be due to the fact that children aged 3–8 years had the lowest rates of seroprotection and served as the main source of infection in the pre-vaccination period. The greater than 90% drop in incidence observed in Tuva in the year following the start of the vaccination campaign was similar to declines observed in two-dose universal vaccination programs in Israel, Uruguay, and the US states that have conducted such vaccination programs [16,17,18]. A similar effect was observed in Brazil following the implementation of single-dose vaccination for toddlers, which led to a drop of the incidence rate from 3.29 to 0.80 per 100,000 between 2014 and 2018. The reduction was observed in all age groups, suggesting a herd immunity effect [19]. Data from the Republic of Korea, where the single-dose vaccination was introduced in the Armed Forces, suggest its high protective efficacy in the adult population. A comparison of the two groups during the vaccine introduction period (2013–2016) revealed a lower incidence rate in the vaccinated group than in the unvaccinated group (0.5 vs. 2.06 per 100,000), resulting in the calculated vaccine effectiveness index to be as high as 95.1% [9].

Evidently, the sharp decline in hepatitis A incidence in Tuva was accounted for by the vaccination program only, as no other interventions such as significant improvements in sanitary conditions or an increase in the availability of quality drinking water occurred in this region. This hypothesis is supported by high incidence rates of other intestinal infections reported in Tuva in the last few years. For example, the incidence of enteroviral infections in Tuva in 2016–2019 was 19.6–174.4 per 100,000 compared to the overall Russian average of 9.8–12.6 per 100,000; the respective shigellosis incidence rates ranged from 49.8 to 119.3 per 100,000 in Tuva, compared to the overall Russian average of 4.6–6.6 per 100,000 [20]. These data suggest the high epidemiological effectiveness of universal single-dose vaccination against hepatitis A, even in the context of a highly endemic environment.

Another important question related to the use of a single dose of HAV vaccine instead of the standard schedule is the duration of seroprotection. Data from Argentina suggest that protective anti-HAV antibody concentrations (≥10 mIU/mL) persist in a majority of vaccinated children up to 9 years of age (median post-vaccination interval 7.7 years) following single-dose vaccination at the age of one year [21]. Our data also demonstrated that up to 91% of children in Tuva have protective concentrations of anti-HAV antibodies five years after single-dose vaccination. The anti-HAV antibody GMC found in this study in children that was detected five years after the vaccination (57.7 mIU/mL) was lower than that reported in Argentina up to five years and nine years after vaccination (122.5 mIU/mL and 170.5 mUI/mL, respectively) [21,22]. These differences may be attributed to a natural boosting effect in some samples tested in Argentina. In our study, samples with anti-HAV antibody concentration > 6000 mIU/mL were excluded from the GMC calculation, as such high titers may be due to HAV exposure or undocumented second-dose booster vaccinations, although the latter were excluded based on participants’ questionnaire data. All in all, five-year monitoring of HAV seroprotection rates in Tuva suggests that antibody titers remained stable within the observation period and the level of seroprevalence in vaccinated children provides protection from incidence flare-ups in the region. Moreover, our data indicate that the single-dose vaccination is an effective tool to control hepatitis A even if it is the only intervention in highly endemic regions, with the lack of significant improvement in sanitary conditions.

The main limitation of the present study relates to the short study period, as we present data only on five-year immunological and epidemiological effectiveness. A further analysis of the incidence rates and antibody levels is needed to understand the duration of the protection following single-dose immunization. The analysis of HAV circulation in Tuva was not within the scope of this research. However, the monitoring of HAV RNA in sewage is very important to access the impact of universal single-dose child vaccination on HAV circulation, and should be the subject of further research.

## 5. Conclusions

Universal single-dose vaccination against hepatitis A in children in a highly endemic region resulted in a rapid and significant decrease in hepatitis A incidence, not only in vaccinated children but also in total population. Protective concentrations of antibodies persist for at least five years following single-dose immunization. These results confirm the effectiveness of single-dose vaccination as a manner of bringing hepatitis A disease prevalence under control in a short period of time. Further monitoring of incidence rates and antibody levels is needed to determine the sustainability of the observed effect, and to establish the need for booster immunization.

## Figures and Tables

**Figure 1 vaccines-08-00780-f001:**
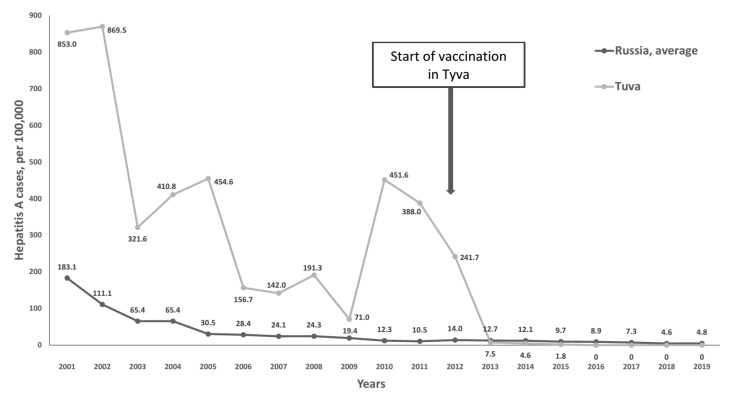
Hepatitis A annual incidence rates in children aged under 18 years in Tuva compared to the average in Russia, 2001–2019.

**Figure 2 vaccines-08-00780-f002:**
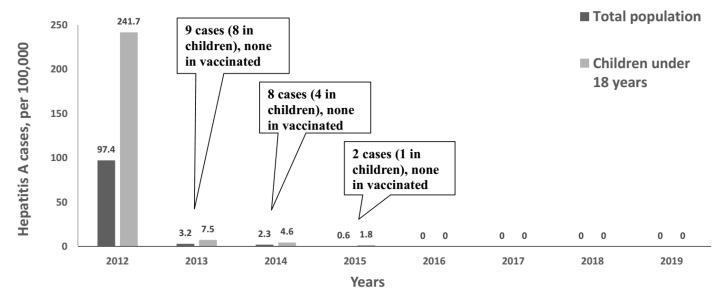
Hepatitis A incidence in both the total population and in children aged under 18 years in Tuva for the period 2012–2019. The callouts indicate the total number of hepatitis A cases registered each year.

**Table 1 vaccines-08-00780-t001:** Cases of hepatitis A reported in Tuva, 2013–2015.

Case #	Age, Years	Sex	Place of Permanent Residence	History of Hepatitis A Vaccination	Infection Risk Factors	Year	Severity of Disease
1	3	Male	Tuva	No	Intrafamilial cluster case (together with cases #2 and #3)	2013	Mild, anicteric
2	16	Female	Tuva	No	Intrafamilial cluster case (together with cases #1 and #3)	2013	Moderate severity, anicteric
3	16	Female	Tuva	No	Intrafamilial cluster case (together with cases #1 and #2)	2013	Moderate severity, anicteric
4	9	Female	Tuva	No	Contact with an infected child in another region in Russia	2013	Moderate severity, icteric
5	10	Female	Tuva	No	No risk factors identified	2013	Moderate severity, icteric
6	12	Male	Tuva	No	No risk factors identified	2013	Moderate severity, icteric
7	12	Female	Tuva	No	No risk factors identified	2013	Mild, anicteric
8	16	Female	Tuva	No	No risk factors identified	2013	Moderate severity, icteric
9	19	Male	Tuva	No	No risk factors identified	2013	Moderate severity, icteric
10	1.5	Male	Tuva	No	Contact with an infected individual in another country	2014	Moderate severity, icteric
11	3	Male	Tuva	No	Contact with an infected individual in another country	2014	Moderate severity, icteric
12	23	Female	Tuva	No	Contact with an infected individual in another country	2014	Moderate severity, icteric
13	35	Female	Tuva	No	No risk factors identified	2014	Subclinical
14	56	Female	Tuva	No	Contact with an infected individual in another country	2014	Subclinical
15	14	Male	Tuva	No	No risk factors identified	2014	Mild, anicteric
16	36	Female	Tuva	No	No risk factors identified	2014	Moderate severity, icteric
17	15	Female	Tuva	No	No risk factors identified	2014	Mild, anicteric
18	34	Male	Tuva	No	No risk factors identified	2015	Moderate severity, icteric
19	13	Male	Tuva	No	No risk factors identified	2015	Moderate severity, icteric

**Table 2 vaccines-08-00780-t002:** Prevalence of anti-HAV IgG in conditionally healthy individuals in Tuva in the pre-vaccination period (in 2008).

Age Group, Years	Number of Participants	Anti-HAV IgG Positive		Proportion of Subjects with a History of Hepatitis A Vaccination (Self-Reported), %
		Number of Positive	% (95% CI)	
<1	88	49	55.7 (45.3–65.6)	0
1–4	100	28	28.0 (20/1–37.5) *	1
5–9	100	42	42.0 (32.8–51.8) *	15
10–14	100	66	66.0 (56.3–74.6) *	8
15–19	100	86	86.0 (77.7–91.6)	5
20–29	100	98	98.0 (92.6–99.9)	5
30–39	100	96	96.0 (89.8–98.8)	3
40–49	100	98	98.0 (92.6–99.9)	2
50–59	100	97	97.0 (91.2–99.4)	4
>60	123	122	99.2 (95.1–100.0)	1.6
All age groups	1011	782	77.3 (74.7–79.8)	4.6

* *p* < 0.001 when compared to older age groups (Fisher’s exact test).

**Table 3 vaccines-08-00780-t003:** Distribution of serum anti-HAV antibody concentrations in samples from vaccinated children collected one month, one year, and five years following single-dose vaccination.

Anti-HAV Antibody Concentration	Detection Rate, % (95% CI)		
	One Month Following Single-Dose Vaccination, n = 451	One Year Following Single-Dose Vaccination, n = 510	Five Years Following Single-Dose Vaccination, n = 463
<10 mIU/ml	2.0% (0.1–3.8%)	6.5% (4.6–9.0%)	8.9% (6.7–11.8%) *
10–19 mIU/ml	0.0% (0.0–1.0%)	0.0% (0.0–0.9%)	17.9% (14.7%–21.7%)
20–6000 mIU/ml	98.0% (96.2–99.0%)	93.5% (91.0–95.4%)	68.9% (64.5%–73.0%) **
>6000 mIU/ml	0.0% (0.0–1.0%)	0.0% (0.0–0.9%)	4.3% (2.8–7.6%)

* *p* < 0.01 when compared to the detection rate in the cohort surveyed one month following vaccination (Fisher’s exact test). ** *p* < 0.01 when compared to the detection rate in cohorts surveyed one month and one year following vaccination (Fisher’s exact test).

**Table 4 vaccines-08-00780-t004:** Serum anti-HAV antibody GMCs in samples from vaccinated children collected one month, one year and five years following single-dose vaccination.

Time Following Single-Dose Vaccination	Number of Samples Used for GMC Calculation *	GMC (95% CI), mIU/mL
1 month	442	40.24 (19.57–60.91)
1 year	477	44.96 (16.09–73.83)
5 years	402	57.73 (16.51–98.95)

* Samples with anti-HAV antibody concentrations <10 mIU/mL and >6000 mIU/mL were excluded from the calculation.

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
