# Peer review of "Universal Single-Dose Vaccination against Hepatitis A in Children in a Region of High Endemicity"

_vaccines, 2020, doi:10.3390/vaccines8040780_

Round 1

Reviewer 1 Report

Submitted manuscript shows the high epidemiological effectiveness of universal single-dose vaccination against hepatitis A, even in the context of a highly endemic environment. While the manuscript is well and clearly written, I have several rather formal concerns.

Seropositivity is defined by antibody levels. The subject “antibody” is missing throughout the manuscript when the levels of anti-HAV response are referred. Thus, instead of “anti-HAV antibody level”, the authors use a deleted version “anti-HAV level”. It is perturbing and should be corrected.

If not published elsewhere, the data describing the vaccination campaign should be presented rather in the introductory part of the Result section than at the end of Introduction of the manuscript (l.70-75).

Age of children cohorts should be expressed as a median (l. 108-111).

Please explain why the seropositivity is defined as antibody levels of ≥10 mIU/m (l. 130)?

Please explain why children under 18 years are shown in Figure 1, while children under 15 in Figure 2

Serologic, demographic, and sanitary conditions (l. 187-202) should be labelled as “not shown”. Please indicate where “the overall average in Russia of 10.6–58.7 per 100,000” (l. 150) is shown.

Author Response

We are very grateful to Reviewer for comments and thorough analysis of our paper.

Comment 1

Seropositivity is defined by antibody levels. The subject “antibody” is missing throughout the manuscript when the levels of anti-HAV response are referred. Thus, instead of “anti-HAV antibody level”, the authors use a deleted version “anti-HAV level”. It is perturbing and should be corrected.

Response 1

We corrected “anti-HAV antibody level” throughout the text in revised manuscript.

Comment 2

If not published elsewhere, the data describing the vaccination campaign should be presented rather in the introductory part of the Result section than at the end of Introduction of the manuscript (l.70-75).

Response 2

This data has not been previously published, so we moved the description of vaccination campaign to the introductory part of the Results section in revised manuscript (page 4, lines 157-160)

 Comment 3

Age of children cohorts should be expressed as a median (l. 108-111).

Response 3

We expressed age of children in study cohorts as a median (page 3, lines 116-118).

 Comment 4

Please explain why the seropositivity is defined as antibody levels of ≥10 mIU/ml (l. 130)?

Response 4

Early studies with HAVRIX® set the threshold of protection between 20–33 mIU/ml using an enzyme-linked immunoassay. This threshold has been cut to 10 mIU/ml in recent years [World Health Organization. The Immunological Basis for Immunization Series Module 18: Hepatitis A. Immunization, Vaccines and Biologicals. 2011. Available at http://apps.who.int/iris/bitstream/10665/44570/1/9789241501422_eng.pdf.]. All studies of the immune response following single-dose HAV vaccine used 10 mIU/ml as cut-off value for seroprotection. Thus, we used the same threshold. We added this point to Methods section to make it clearer (page 3, lines 138-139).

Comment 5

Please explain why children under 18 years are shown in Figure 1, while children under 15 in Figure 2

Response 5

Registered incidence of infectious diseases in Russia is reported in Russia for three categories of people: the total population, children under 15 years of age, and children under 18 years of age. Figure 1 shows the incidence in children under 18 years, because incidence rates were the highest in that age group in Tuva in pre-vaccination period. We opted to show incidence in children under 15 years in Figure 2, as this age group is closer to vaccinated cohort (3-8 years).

To avoid any confusion, we changed Figure 2 in revised manuscript, now it shows incidence in children under 18 years.  This did not distort the meaning of the figure, because all these age groups overlap indeed. The corresponding text was also corrected (page 5, lines 179-182). The respective changes have been made in Abstract (page 1, lines 25-26). 

Comment 6

Serologic, demographic, and sanitary conditions (l. 187-202) should be labelled as “not shown”.

Response 6

We indicated in revised manuscript that these data are not shown at the end of the respective paragraph (page 7, line 211).  

Comment 7

Please indicate where “the overall average in Russia of 10.6–58.7 per 100,000” (l. 150) is shown.

Response 7

These indicators are given in the text by mistake (they corresponded to total population, indeed). In the revised version, the corrected indicators in the text correspond to those shown in Figure 1, 7.5-183.1 per 100,000 (page 4, line 165).

Reviewer 2 Report

In the manuscript, the authors reported a large-scale survey on the efficacy of single-dose hepatitis A virus (HAV) vaccination in Tuva, Russia from 2012 to 2019. The immunization outcome turned out to be encouraging in protection efficacy and antibody persistency. HAV infection rate dropped significantly by the vaccination program. The study may provide a positive example for many countries in which the geographic and economic situations are similar to Tuva. 

A minor revision is needed in the conclusion: "Universal single-dose vaccination against hepatitis A in children in a highly endemic region
resulted in a rapid and significant decrease in hepatitis A ncidence, not only in vaccinated children but also in unvaccinated adolescents and adults." (line 282 - 284) . There is no supporting datum for the idea that "a rapid and significant decrease of HAV in unvaccinated adolescents and adults" in the manuscript, hence the statement is not adequate. 

"Since 2016, not a single case of hepatitis A has been reported in Tuva." (line 24), it is "no" instead of "not".  

Author Response

We are grateful to the Reviewer for the comments and thorough analysis of the manuscript.

Comment 1

A minor revision is needed in the conclusion: "Universal single-dose vaccination against hepatitis A in children in a highly endemic region resulted in a rapid and significant decrease in hepatitis A incidence, not only in vaccinated children but also in unvaccinated adolescents and adults." (line 282 - 284). There is no supporting datum for the idea that "a rapid and significant decrease of HAV in unvaccinated adolescents and adults" in the manuscript, hence the statement is not adequate.

Response 1

We change this statement to the following: “Universal single-dose vaccination against hepatitis A in children in a highly endemic region resulted in a rapid and significant decrease in hepatitis A incidence, not only in vaccinated children but also total population.” (page 9, lines 323-325). This statement was supported our data on hepatitis A incidence in Tuva. 

Comment 2

"Since 2016, not a single case of hepatitis A has been reported in Tuva." (line 24), it is "no" instead of "not". 

Response 2

We rephrased this sentence and changed “not a single case” to “no cases” (page 1, line 27 in revised manuscript).  

Reviewer 3 Report

Thank you very much for allowing me to review the article titled “UNIVERSAL SINGLE-DOSE VACCINATION AGAINST HEPATITIS A IN CHILDREN IN A REGION OF HIGH ENDEMICITY” (vaccines-1027765).

Hepatitis A is a disease preventable by vaccination, therefor universal pediatric vaccination programs may be beneficial in regions with endemicity of hepatitis A. This vaccination can be on two doses and of single-dose, this vaccination against hepatitis A with one dose was first implemented in toddlers in Argentina in 2005 and its recommended by the WHO.

This article inform about the experience of this vaccination on 65,097 children who have received single-dose immunization in Tuva (Russia).

Summary: it must indicate the objective of the work and the methodology, so that it is informative for the reader of the article.

Introduction: At the end of the introduction you must indicate the objective of the study, please rewrite line 78-79.

Material and methods: Please specify the design of the study, does it appear to be a study before and after an intervention or is it a historical cohort study? This section indicates that the sample is 1,001 healthy individuals from Tuva of Ten age groups. These data do not coincide with the title of the work that specifies that it is a study in children, and neither with the abstract that indicates 65,097 children, please clarify this controversy.

Presents approval of the Ethics Committee.

 Indicate the tests applied and the confidence interval calculated.

 Results: The results of the total population are presented, children under 15 years of age, and children under 18 years, this is a bit strange since those under 18 are included all those under 15, these groups are superimposed, how do they justify it?

 Table 2 is for the entire population, this does not coincide with the job title.

Table 3 and 4 of which population are the data of serum anti-HAV concentrations being presented? Please specify….

Discussion: It is very brief, it should be expanded, including other studies, strengths of this study and limitations.

Author Response

We are very grateful to Reviewer for comments and thorough analysis of our paper.

Comment 1

Summary: it must indicate the objective of the work and the methodology, so that it is informative for the reader of the article.

Response 1

We indicated the objective and methodology of our study in revised manuscript as “The objective of this cross-sectional study was the assessment of the immunological and epidemiological effectiveness of vaccination program five years following its implementation” (page 1, lines 20-22).

Comment 2

Introduction: At the end of the introduction you must indicate the objective of the study, please rewrite line 78-79.

Response 2

We indicated the objective of the study and also added the reason for the presenting the data on historical cohort (page 2, lines 81-85)

Comment 3

Material and methods: Please specify the design of the study, does it appear to be a study before and after an intervention or is it a historical cohort study? This section indicates that the sample is 1,001 healthy individuals from Tuva of Ten age groups. These data do not coincide with the title of the work that specifies that it is a study in children, and neither with the abstract that indicates 65,097 children, please clarify this controversy.

Response 3

Our study includes the analysis of changes in hepatitis A incidence following the implementation of single-dose vaccination and the five-year immunogenicity data. We also included the data on herd immunity to HAV in Tuva in pre-vaccinated period, obtained on the historical cohort. We believe that these data are useful as it demonstrates the epidemiological background and the proportion of individuals susceptible to HAV in different age groups. We gave the rationale for inclusion of these data in Introduction section (page 2, lines 84-85) and in the beginning of the subsection 2.2. Study cohorts (page 3, line 100). We also discuss these data in the expanded Discussion section, pointing that “in a cohort sampled in 2008, almost 100% of healthy individuals over the age of 20 years were seropositive, suggesting long-term intensive HAV circulation in this region. The relatively low seroprotection rates observed during the pre-vaccination period in children aged 1–14 years may explain why the majority of cases reported in Tuva were reported in children and adolescents.” (page 8, lines 261-265). 

Comment 4

Presents approval of the Ethics Committee.

Response 4

Information about approval of the Ethics Committee is given in subsection 2.2. Study cohorts (page 3, lines 110-111 and lines 124-125).

Comment 5

Indicate the tests applied and the confidence interval calculated.

Response 5

Detailed information about tests used for anti-HAV antibody detection and quantification is given in subsection 2.3. Anti-HAV testing (page 3, lines 135-136 and page 4, lines 145-146). For statistical analysis we used Fisher exact test (page 4, lines 152-153). All calculated 95% confidence intervals are given in tables 2, 3, and 4).

Comment 6

Results: The results of the total population are presented, children under 15 years of age, and children under 18 years, this is a bit strange since those under 18 are included all those under 15, these groups are superimposed, how do they justify it?

Response 6

Registered incidence of infectious diseases in Russia is reported in Russia for three categories of people: the total population, children under 15 years of age, and children under 18 years of age. Figure 1 shows the incidence in children under 18 years, because incidence rates were the highest in that age group in Tuva in pre-vaccination period. We opted to show incidence in children under 15 years in Figure 2, as this age group was closer to vaccinated cohort (3-8 years). To avoid any confusion, we changed Figure 2 in revised manuscript, now it shows incidence in children under 18 years.  This did not distort the meaning of the figure, because all these age groups overlap indeed. The corresponding text was also corrected (page 5, lines 179-182).  The respective changes have been made in Abstract (page 1, lines 25-26). 

Comment 7

Table 2 is for the entire population, this does not coincide with the job title.

Response 7

We changed the title of the table 2 to “Prevalence of anti-HAV IgG in conditionally healthy individuals in Tuva in pre-vaccination period (2008)” to make it clearer. The rationale for including these data into paper are given in response to Comment 3.

Comment 8

Table 3 and 4 of which population are the data of serum anti-HAV concentrations being presented? Please specify….

Response 8

We specified in titles of tables 3 and 4 that these samples obtained from vaccinated children

Comment 9

Discussion: It is very brief, it should be expanded, including other studies, strengths of this study and limitations.

Response 9

We expanded Discussion section, added data on immunogenicity and effectiveness of single-dose vaccination against hepatitis A (page 8, lines 247-250, and page 9, lines 279-286).

We believe that the strength of our study is that our data indicate that the single-dose vaccination is an effective tool to control hepatitis A even if it is the only intervention in highly endemic region, with the lack of significant improvement in sanitary conditions. This data may provide an example for many countries with similar economic and epidemic situation.

The main limitation of our study relates to the short study period, as we present here only data on five-year immunological and epidemiological effectiveness. Further analysis of incidence rates and antibody levels is needed to understand the duration of the protection following single-dose immunization. The analysis of HAV circulation in Tuva was out of our attention. However, the monitoring of HAV RNA in sewage is very important to access the impact of universal single-dose child vaccination on HAV circulation and is the subject of our further research. We added the respective paragraph to the Discussion section (page 9, lines 312-320).

Round 2

Reviewer 3 Report

After reviewing the new version of the manuscript "“UNIVERSAL SINGLE-DOSE VACCINATION AGAINST HEPATITIS A IN CHILDREN IN A REGION OF HIGH ENDEMICITY” (vaccines-1027765)." and the response of its authors, I have verified that the authors have made the clarifications indicated.

This article provides the experience if a universal single-dose vaccination against hepatitis A in children in a highly endemic region and they found as resulted in a rapid and significant decrease in hepatitis A incidence, not only in vaccinated children but also in total population. These results confirm the effectiveness of single-dose vaccination as an option in Public Health